# Cultivable Endophyte Resources in Medicinal Plants and Effects on Hosts

**DOI:** 10.3390/life13081695

**Published:** 2023-08-06

**Authors:** Yiming Wang, Yongjing Zhang, Hao Cong, Changgen Li, Jiaying Wu, Ludan Li, Jihong Jiang, Xiaoying Cao

**Affiliations:** The Key Laboratory of Biotechnology for Medicinal and Edible Plant Resources of Jiangsu Province, School of Life Sciences, Jiangsu Normal University, Xuzhou 221116, China; 1020210013@jsnu.edu.cn (Y.W.); 1020210011@jsnu.edu.cn (Y.Z.); conghao@jsnu.edu.cn (H.C.); 1020200018@jsnu.edu.cn (C.L.); 1020220016@jsnu.edu.cn (J.W.); lild@jsnu.edu.cn (L.L.); jhjiang@jsnu.edu.cn (J.J.)

**Keywords:** medicinal plants, endophyte, biocontrol, medicinal components

## Abstract

With the increasing demand for medicinal plants and the increasing shortage of resources, improving the quality and yield of medicinal plants and making more effective use of medicinal plants has become an urgent problem to be solved. During the growth of medicinal plants, various adversities can lead to nutrient loss and yield decline. Using traditional chemical pesticides to control the stress resistance of plants will cause serious pollution to the environment and even endanger human health. Therefore, it is necessary to find suitable pesticide substitutes from natural ingredients. As an important part of the microecology of medicinal plants, endophytes can promote the growth of medicinal plants, improve the stress tolerance of hosts, and promote the accumulation of active components of hosts. Endophytes have a more positive and direct impact on the host and can metabolize rich medicinal ingredients, so researchers pay attention to them. This paper reviews the research in the past five years, aiming to provide ideas for improving the quality of medicinal plants, developing more microbial resources, exploring more medicinal natural products, and providing help for the development of research on medicinal plants and endophytes.

## 1. Introduction

Medicinal plants refer to plants used in medicine to prevent and treat diseases [1]. All or part of medicinal plants are used in medicine and will also be used as raw materials for the pharmaceutical industry, which have a wide range of medicinal and economic uses [2]. Especially in the field of traditional medicine represented by traditional Chinese medicine and Indian folk medicine, medicinal plants, as the main source of natural drugs, provide very important health care services for the population of developing countries [3,4,5]. With the rapid development of modern medicine, many clinical drugs still come from natural products extracted from medicinal plants [2]. Although many kinds of medicinal plants have been used in clinical treatment, due to environmental stress, overexploitation, low reproductive capacity, and other factors, some rare, high-demand, and wild medicinal plant resources cannot meet the market demand, so how to improve the germplasm resources of medicinal plants has become an urgent problem to be solved.

In recent years, researchers have gradually realized that endophytes can play an important role in affecting the quality and yield of medicinal plants through special microbe-plant interactions [6,7]. Plant endophytes are microbial groups that widely exist in healthy medicinal plant tissues, coexist harmoniously with host plants, and do not cause significant damage to hosts [8]. They are also an important part of theplant micro-ecosystem, which is rich in species, mainly including endophytic fungi, endophytic bacteria, and endophytic actinomycetes [7,9,10]. At present, endophytes have been isolated from a variety of medicinal plants, and many endophytes have been verified to secrete plant hormones, growth factors, etc., which are conducive to plant growth and development and can also regulate the accumulation and production of active ingredients in medicinal plants [10,11]. They increase the active ingredients of the host by producing the same or similar active products as the active ingredients in the host [11,12,13]. The most interesting property of endophytes is that they can convert the original active ingredients of the plant into new compounds. In 1993, Stierle et al. isolated an endophytic fungus from *Taxus brevifolia* and found that it can produce paclitaxel, an anti-tumor substance similar to the host plant, which inspired researchers to find bioactive substances from endophytes of medicinal plants [14]. Endophytes provide more resources of new bioactive metabolites, especially alkaloids, saponins, quinones, flavonoids, terpenoids, etc., which have a lot of biological activities and have also become research hotspots in the composition and production of natural drugs [8].

In order to improve the quality of medicinal plants, more needs to be known about the special relationship between endophytes and medicinal plants. In this review, Pubmed and Endnote were used to explore articles using the keywords ‘medicinal plants’, ‘endophyte’, ‘metabolites’, ‘growth promotion’, ‘stress resistance’, and ‘Biocontrol’ and summarize the research on the function of culturable endophytes of medicinal plants in the past five years (2019–2023). The development and utilization of endophytes in medicinal plants were prospected to provide references for the development of endophyte products and improving the quality of medicinal plants.

## 2. Medicinal Plants and Their Cultivable Endophyte Resources

Endophytes in medicinal plants mainly include endophytic fungi, endophytic bacteria, and endophytic actinomycetes, and they are rich in species diversity [6,7]. It is found that the biological functions of these endophytes have a great influence on medicinal plants [10,11], so obtaining more microbial resources, especially those with biological activity, can greatly promote the development of the medicinal plant industry.

### 2.1. Culturable Endophytic Bacteria Diversity in Medicinal Plants

*Atractylodes macrocephala* Koidz., called Baizhu in Chinese, is a medicinal plant used in traditional Chinese medicine theoretical systems to treat gastrointestinal dysfunction, cancer, osteoporosis, obesity and other symptoms, and has various pharmacological activities [15]. Wu et al. [16] explored the cultivable endophytic bacteria in the stems, leaves, roots, and rhizomes of *Atractylodes macrocephala* Koidz. in four different regions and their potential correlation with plant bioactives. A total of 118 endophytic bacteria belonging to 3 phyla, 5 classes, 11 orders, 26 families and 48 genera were identified from four *Atractylodes macrocephala* Koidz. tissues. Among them, *Bacillus* sp. is the most widely distributed. Dendrobium is one of the largest genera of *Orchidaceae*, with more than 1500 species distributed all over the world [17]. As a medicinal plant, dendrobium has greatly. contributed to the medical industry with its anticancer, antifatigue, and gastrointestinal protective effects [18]. In addition, there are also many microbial resources in dendrobium. Wang et al. [19] isolated and cultured endophytic bacteria from *Dendrobium officinale* samples of six different sources and cultivars. A total of 165 cultivable endophytic bacteria were isolated from sterilized *Dendrobium officinale* stems and classified into 43 species based on 16S rRNA gene sequence analysis, of which 14 strains had anti-plant-pathogenic activity. Mulberry, which belongs to the genus *Morus* of the *Moraceae* family, is an aggregated berry that is oval-shaped, palatable, and also rich in nutrients; it is regarded as a very important medicinal and edible plant due to its rich, effective chemical composition and wide range of biological activities [20,21,22]. Xu et al. [23] isolated a total of 608 endophytic bacteria from four mulberry cultivars, belonging to 4 phyla and 36 genera.

Bacteria, as the largest group of plant endophytes, have been isolated from many kinds of medicinal plants and widely studied due to their biocontrol functions [24,25,26,27,28,29,30,31,32]. By reviewing the recent literature on most of the endophytic bacteria of medicinal plants including *Bacillus* sp., *Pseudomonas* sp., *Enterobacter* sp., *Agrobacterium* sp., etc., and a large number of endophytic bacteria in the roots, stems, and leaves, we collated some of the relevant data of the isolated endophytic bacteria in Table 1.

### 2.2. Culturable Endophytic Fungal Diversity in Medicinal Plants

*Aconitum heterophyllum* is an alkaloid-rich medicinal plant which is widely used in traditional Chinese medicine clinics [38,39]. A total of 328 fungal isolates were found in leaf, stem and root tissues of plants by Hafeez et al., and 12 endophytic fungal species were identified by molecular characterization [39]. *Crocus sativus* L. (family Iridaceae) has been widely used as an antimicrobial, antidepressant, digestive, anticancer, and anticonvulsant medicine due to its abundant natural products as well as antioxidant activity [40,41]. Lu et al. [42] isolated endophytic fungi from five different locations in *Crocus sativus* tissues (corm, scape, leaf, petal, and stigma) and identified a total of 32 endophytic fungal groups, assigned to seven orders within four classes. Wang et al. [43] isolated 34 endophytic fungi from *Salvia miltiorrhiza*, a traditional Chinese medicine, belonging to 10 genera and 16 species, and *Epicoccus* sp. SX19 and *Colletotrichum gloeosporioids* showed strong inhibitory effects on five pathogens. Ogbe et al. [30] isolated a total of 11 endophytic fungi from the roots and leaves of a drought tolerant mint species *Endostemon obtusifolius*. Similarly, five endophytic fungi were isolated from the leaf segments of wild *Dendrobium* nobile and identified as *Colletotrichum tropicicola*, *Fusarium keratoplasticum*, *Fusarium oxysporum*, *Fusarium solani*, and *Trichoderma longibrachiatum* [44]. *Codonopsis pilosula*, as a famous medicinal and food homologous plant, has functions such as strengthening the spleen, tonifying the lungs, and engendering liquid in traditional Chinese medicine [45]. Fan et al. [46] obtained 205 strains of endophytic fungi from the roots of *Codonopsis pilosula*, collected from six regions in Gansu Province, China, of which *Fusarium* sp., *Aspergillus* sp., *Alternaria* sp., *Penicillium* sp., and *Plectosphaerella* sp. were the dominant genera. *Vernonia anthelmintica* (L.) Willd has a long history in the treatment of several diseases related to skin, central nervous system, kidney, gynecology, gastrointestinal, metabolism, and general health [47]. Researchers have isolated more than 30 types of endophytic fungi from *Vernonia anthelmintica.* [48]

From the research in recent years, it can be seen that *Fusarium* sp., *Aspergillus* sp., and *Penicillium* sp. can be isolated from most medicinal plants, and because of their many biological functions, they are regarded as the key research objects of endophytic fungi in medicinal plants. The recent research results are summarized in Table 2.

### 2.3. Culturable Endophytic Actinomycetes Diversity in Medicinal Plants

*Dioscorea* has powerful medicinal functions and is a potential source of bioactive substances for combating various diseases [49]. Zhou et al. [50] isolated 116 actinomycetes from the tissues of *Dioscorea opposita* Thunb. and found a new *Streptomyces* sp. with strong biocontrol function. As a traditional Chinese medicine, *Eucommia ulmoides* Oliv. has been used to treat various diseases since ancient times [51]. The research group led by Mo et al. [52,53] isolated two new species of *Nocardia* sp. from the leaves and roots of *Eucommia ulmoides* Oliv. *Thymus roseus* schipcz is one of the traditional Chinese herbs belonging to Lamiaceae and has been proven to have anti-inflammatory, antioxidant, anti-cancer and other functions [54]. Musa et al. [54] isolated 128 strains from the roots, stems and leaves of *Thymus roseus* schipcz, with a predominance of *Streptomyces* sp., followed by *Nocardiopsis* sp., *Micrococcus* sp., *Kocuria* sp., and others. *Viola odorata* grows in the high altitude area of the Himalayas and is used as a natural medicine because of its antidiabetes, anti-inflammatory, and other functions [55,56]. Salwan et al. [56] isolated a Streptomyces strain with antioxidant and antibacterial activity from the medicinal plant *Viola odorata* collected in the Himalayas, which has the potential to produce antibacterial and antioxidant components. *Xanthium sibiricum* is a well-known Chinese herbal medicine commonly used to treat autoimmune and inflammatory diseases [57]. Hu et al. [58] isolated two new Streptomyces strains from healthy leaves and seeds of *Xanthium sibiricum*.

According to the research on endophytic actinomycetes of medicinal plants in recent years, actinomycetes are mainly distributed in the roots of medicinal plants, and their number is greater than that in other tissues of plants. *Streptomyces* sp. is the main research object of actinomycetes, and *Streptomyces* sp. has received extensive attention because of its strong biological activity [59]. Actinomycetes of other genera can also be isolated from medicinal plants, but the number is relatively small compared with Streptomyces. Many studies have isolated new species of bioactive endophytic actinomycetes from medicinal plants, which greatly expanded microbial resources and laid a foundation for the industrial application of actinomycetes [52,53,58,60,61]. The results of endophyte isolation from medicinal plants in recent years are shown in Table 3.

## 3. Beneficial Effects of Endophytes from Medicinal Plants on the Host

In recent years, many studies have shown that endophytes have made important contributions in promoting the growth of medicinal plants, enhancing the stress tolerance of medicinal plants and biological control of plant diseases.

### 3.1. Promoting the Growth of Medicinal Plants

Khan et al. [65] isolated an endophytic fungus *Acremonium* sp. Ld-03, which has antibacterial activity and can produce indoleacetic acid and siderophores, from the medicinal plant *Lilium davidii*. After diluting the culture medium to different concentrations and conducting liquid culture on the host, it was found that with the application of 40% culture dilution of *Acremonium* sp., the root and bud length of *Allium tuberosum* can be significantly increased. Zou et al. [66] isolated a *Bacillus subtilis* strain from the medicinal plant *Aconitum carmichaelii* DEBX., which can produce gluconase, cellulase, protease, indole acetic acid, siderophore, antifungal lipopeptides, and polyketides and significantly increase the fresh weight and dry weight of the host stem, main root, and lateral root. Tao et al. [67] isolated four strains of endophytic bacteria with indole acetic acid production, phosphate solubilization, and nitrogen fixation abilities from the precious traditional Chinese medicine *Pairs polyphylla* var. *yunnanensis* and significantly increased the host’s biomass. Mathur et al. [68] isolated *Aspergillus niger* which can produce gibberellin from *Albizia lebbeck* (L.) Benth, effectively promoting the seed germination of wheat, barley, and millet. Purushotham et al. [69] isolated an endophytic actinomycete *Nocardia* sp. TP1BA1B from the native medicinal plant Pseudowitera colorata (horopito) in New Zealand, which has the function of dissolving phosphate, producing siderophores, and promoting the growth of host seedlings.

The growth of medicinal plants is related to various factors, such as light, temperature and microorganisms, among which endophytes play a crucial role in the host’s growth process [70]. Endophytes promote the growth and development of plants in different ways, such as secreting siderophore to improve the utilization rate of iron in plants. Endophytes with nitrogen fixation, phosphate solubilization, and potassium solubilization abilities promote the growth of medicinal plants by promoting the absorption of nitrogen, phosphorus, and potassium. Endophytes can also promote plant growth by providing growth hormone to the host, such as indole-3-acetic acid, indole-3-acetonitrile, gibberellin, and cytokinins [10,71,72].

### 3.2. Enhance the Stress Tolerance of Medicinal Plants

Li et al. [73] isolated a strain of *Streptomyces* from *Glycyrrhiza uralensis* and confirmed through inoculation that this strain can enhance the tolerance of the host to drought, salt, and drought salt conditions. Studies have found that under drought stress, the growth of *Helianthus tuberosus* L. (*Jerusalem artichoke*) can be better promoted by the inoculation of endophytic bacteria [74]. *Sphingomonas paucimobilis*, an endophytic bacterium in the rare medicinal plant *Dendrobium officinale*, has good resistance to stresses of salt, drop and cadmium, and this strain is the only one with growth promoting ability reported in this species [75]. Some researchers isolated Endophytes from *Astragalus mongholicus* and co-inoculated them with *Trichoderma* strains under drought conditions, which significantly improved the root biomass, root length, calycosin-7-O-β-D-glucoside content of the host and activities of nitrate reductase and soil urease [76].

Endophytes can enhance the environmental adaptability of host plants by enhancing the expression of stress resistance related genes in host plants and increasing the activity of related enzymes [77]. In addition, some endophytes may also produce antibiotic compounds, antimicrobial peptides, or alkaloids to help the host resist pests and diseases [78,79].

### 3.3. Promoting the Accumulation of Secondary Metabolites in Medicinal Plants

*Salvia miltiorrhiza* is widely used in East Asia because of its anti-tumor, anti-inflammatory, and cardiovascular protection, and tanshinone and salvianolic acid are important medicinal components [80,81]. Endophytic fungus *Cladosporium tenuissimum* DF11 isolated from *Salvia miltiorrhiza* by Chen et al. [80] promoted the biosynthesis and accumulation of tanshinone in roots by upregulating the expression of HMGR, DXS, DXR, GGPPS, CPS, KSL and CYP76AH1, which are key enzyme genes of the tanshinone biosynthesis pathway. Other researchers used endophytes of *Salvia miltiorrhiza* to prepare elicitors, which affected the accumulation of metabolites in hairy roots of *Salvia miltiorrhiza* by inducing the expression of key genes (SmAACT, SmGGPPS, and SmPAL) [81]. Researchers have also isolated two strains of fungi from *Salvia abrotanoides* that can increase host cryptotanshinone and tanshinone IIA production [82]. Ye et al. [83] isolated three Endophytes from *Houttuynia cordata*, named *Ilyonectria liriodendra*, unidentified fungal sp., and *Penicillium citrinum*, which can respectively increase the phenolic compounds of the host, increase the components such as afzelin, decanal, and 2-undecanone, and increase the biomass of the host. Xie et al. [84] isolated an endophytic fungus, *Schizophyllum commune* from *Panax ginseng* and significantly enhanced the expression of key enzyme genes involved in ginsenoside biosynthesis pathways such as pgHMGR, pgSS, pgSE, and pgSD under co-culture conditions, promoting the accumulation of specific ginsenosides.

Endophytes can directly participate in the synthesis of secondary metabolites of medicinal plants and can also induce the formation of secondary metabolites of medicinal plants [85]. Selecting appropriate endophytes to act on medicinal plants can improve the content of secondary metabolites, which is of great significance for improving the quality of medicinal plants, protecting endangered medicinal plants, and synthesizing and developing new drugs.

### 3.4. Helping the Host Resist Pathogens

*Streptomyces dioscori* isolated from *Glycyrrhiza uralensis* exhibited inhibitory effects on three pathogenic fungi: *Rhizoctonia solani*, *Fusarium acuminatum*, and *Sclerotinia scrotiorum* [73]. An endophytic fungus, *Diaporthe* sp., was isolated from the leaves of the Indian medicinal plant *Chloranthus elator* Sw., and its camphor odor volatiles showed inhibitory effects on eight fungal pathogens in vitro [86]. *Burkholderia gladioli*, an endophytic bacterium from *Crocus sativus* Linn., can reduce corm rot and increase endogenous jasmonic acid (JA) level and expression of JA-regulated and other plant defense genes through antibacterial effects and improve the host’s resistance to *Fusarium oxysporum* [87].

Many endophytes can inhibit the occurrence of plant diseases caused by pathogenic bacteria. Endophytes can inhibit the activity of pathogens by inducing host resistance to resist the infection of pathogens and competing with pathogenic bacteria to produce antibiotics, hydrolytic enzymes, alkaloids, and other secondary metabolites and signal interference to resist disease caused by pathogens in host plants [88].

## 4. Medicinal Components Produced by Endophytes in Medicinal Plants

An endophytic fungus, *Xylaria feejeensis*, was derived from the medicinal plant *Geophila repens* in Sri Lanka. Integrated acids, derived from fungal metabolites, have strong antibacterial activity and are a potential resource of antibiotics [89]. Saikosaponin d (SSd) is an important medicinal component of the medicinal plant *Bupleurum scorzonerifolium* Willd. Some researchers isolated two endophytic Fungi from *Bupleurum scorzonerifolium* Willd., which can produce saikosaponin through UPLC/Q-TOF-MS detection [90]. The metabolite 7-methoxy-13-dehydroxypaxilline of *Penicillium* sp., an endophytic fungus isolated from the leaves of the traditional medicinal plant *Baphicacanthus cusia* (Nees) bremek., is a new indole diterpenoid, which exhibits anticancer activity [91]. Gu et al. [92] isolated a new compound, phomopolide G, from the fermentation broth of endophytic fungi of *Artemisia argyi*, which showed a wide range of antibacterial activities. *Cochliobolus* sp., an endophytic fungus of the Indian medical herb *Andrographis paniculata*, can metabolize the alkaloid aziridine, 1-(2-aminoethyl) and can be antibacterial and insect resistant [93]. A new crystalline compound 5-(1-hydroxybutyl)-4-methoxy-3-methyl-2h-pyran-2-one (c-hmmp) in the endophytic fungus *Colletotrichum acutatum* in the medicinal plant *Angelica sinensis* shows antibacterial, antimalarial, anticancer, antioxidant, and other activities [94]. The antibacterial compound 1,4-dihydroxy-2-methyl-anthraquinone was also isolated from the endophytic bacteria of *Archidendron pauciflorum* [32]. The structures of the above compounds are shown in Figure 1.

At present, many alkaloids, flavonoids, phenolic acids, terpenoids, coumarins, and other substances with antioxidant activity have been isolated from endophytes and their secondary metabolites of medicinal plants [10,11,32,95,96,97]. These natural antioxidant active substances often have anti-inflammatory, antioxidant, antibacterial, anti-tumor, antiviral and other functions [10,11,98]. Through searching the literature in the past five years, it was found that the bioactive natural products of endophytes in medicinal plants were metabolized by fungi, while the metabolites of bacteria and actinomycetes were mainly antibiotics [10]. More and more studies have found that natural product produced by host plants may be produced by endophytes or metabolites closely related to endophytes [99]. Further research on the metabolites of endophytes will be of great significance for the development of medicinal plants and clinical drugs.

## 5. Discussion

In recent years, people have gradually realized that endophytes play an important role in affecting the yield and quality of crude drugs by interacting with the host in a specific way. The traditional method of endophyte research is to use the artificial medium to culture, isolate, and purify microorganisms to obtain pure culture strains and use microscopic technology to observe and classify their morphology. According to the physiological and biochemical characteristics of the strain, 16S rRNA, its gene sequencing, and other molecular biological methods were used for gene identification [10].

The rapid development of gene sequencing technology, especially the emergence of high-throughput sequencing and other technical means, has brought unprecedented development to microbiology research [100]. High throughput sequencing technology has been applied to the study of the structure and diversity of a variety of plant endophytes and avoids the process of endophyte culturing in order to explore more microbial resources [100,101,102,103]. High throughput sequencing technology can explore the richness of microbial resources in medicinal plants, but only a small part can be isolated, which also means that there is still a great research potential for endophytes in medicinal plants. Therefore, it is necessary to optimize cultivation methods to obtain more cultivable microbial resources. In the process of endophytes’ isolation, the method of surface disinfection and the formula of the medium will affect the isolation of endophytes [104,105,106]. Excessive disinfection can cause damage to the endophytes of the plant, while incomplete disinfection can lead to contamination by other microbes [104,105]. Different carbon sources, nitrogen sources, and nutrients can also cause great differences in the isolation results of endophytes. The use of antibiotics in the culture medium can effectively improve the isolation efficiency of endophytes [104]. Some researchers have also added plant extracts to the culture medium to greatly increase the number of culturable endophytes [107].

Medicinal plants are the foundation of the development of the pharmaceutical industry. With the understanding and utilization of the cultivation, growth, and various functions of medicinal plants, the quality of medicinal plants has attracted great attention in society [108]. The quality and yield of medicinal plant raw materials are largely affected by many factors, such as plant genetic background, ecological habitat of plants, and soil nutrients [109,110]. Endophytes live in medicinal plants and have the functions of promoting the growth of host plants, enhancing the stress resistance of host plants, and regulating the synthesis of secondary metabolites and being able to metabolize medicinal compounds [111], as shown in Figure 2. Utilization of endophytes can reduce the use of fertilizers and pesticides, which plays an important role in protecting the environment [112,113].

The secondary metabolites of endophytes from most medicinal plants have medicinal activities, which have great potential in the development of new drugs [11,113]. If endophytes of medicinal plants can be used to produce drugs in the future, it will make up for the demand for some rare medicinal plants to a certain extent [114]. In the case of taxol, as a diterpene with anticancer activity isolated from *Taxus chinensis*, the content of taxol in *Taxus chinensis* is very low, and its synthesis is complicated [66,115]. The slow growth rate and scarce resources of *Taxus chinensis* also limit the application of taxol. Since the isolation of taxol producing endophytes from *Taxus chinensis*, a new pathway for taxol production has been developed [116,117,118,119]. The endophytes, especially endophytic funji of *Taxus chinensis* have high growth rate, low cultivation cost, and are not affected by climate change [116]. Therefore, they have excellent prospects as taxol producers. The research methods of secondary metabolites of endophytes in plants are usually based on the isolation and purification strategies of natural products after fermentation of endophytes [120]. However, when endophytes leave the host and are cultured for multiple generations, the down-regulation of synthetic coding genes results in decreased metabolic capacity and biological functions [116,121,122]. This also causes losses to the benefits of using endophytic fungi to produce paclitaxel in industry. Fortunately, these problems can be overcome by optimizing culture methods, co-culturing, silencing rate-limiting genes and transcription factors, and activating the taxol synthesis gene cluster in fungi [115,116]. Therefore, optimizing the culture method of endophytes, promoting the metabolism of endophytes of medicinal plants, producing more metabolites with medicinal effects, and maintaining the biological function of endophytes are also urgent problems to be solved. Similarly, for other rare medicinal plants, we can also find endophytes with the potential of metabolizing active products, and through optimization of cultivation methods, genetic optimization and other methods, replace the rare medicinal plants to become producers of natural product and allow endophytes to maintain their biological functions, which is research with great potential and significance.

The use of chemical pesticides cause serious pollution to the environment and even endangers human health [123], so it is necessary to put forward more sustainable development strategies for the environment and human health and find suitable pesticide substitutes. Endophytes come from plants and act on plants, and they are rich in species and have strong biological control functions [67,124]. Therefore, they are the most suitable to replace chemical pesticides in the cultivation and protection of medicinal plants, and are an important environmental protection strategy. The metabolites of endophytes in medicinal plants are also a huge treasure house for the discovery of medicinal ingredients, which can greatly make up for the shortage of natural resources, which also makes the research of endophytes attract more researchers’ attention [8,125]. In order to fully develop the research and application of endophytes in medicinal plants, the following problems need to be solved urgently: 1. How to make the biocontrol strains survive in the environment outside the plant for a long time? 2. Can the reduced metabolic capacity and biological function of endophytes after multi-generation culturing be overcome by optimizing culture regulation? 3. In vitro endophytes can produce a variety of secondary metabolites to antagonize pathogenic bacteria and inhibit their growth. Can the development of new green pesticides be produced in large quantities with high efficiency? 4. How can we improve the fermentation efficiency of endophytes in medicinal plants to tap more abundant potential medicinal ingredients? To solve the above problems, exploring the interaction mechanism between endophytic bacteria and hosts as well as the metabolic mechanism of medicinal components and conducting more research on the application of endophytes in production will be of great help to the production of medicinal plants and the development of medicinal components.

## Figures and Tables

**Figure 1 life-13-01695-f001:**
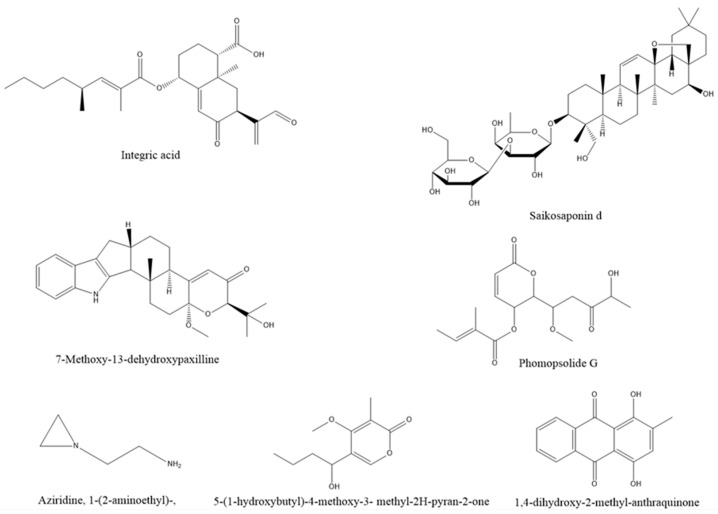
Some active ingredients from endophytes of medicinal plants.

**Figure 2 life-13-01695-f002:**
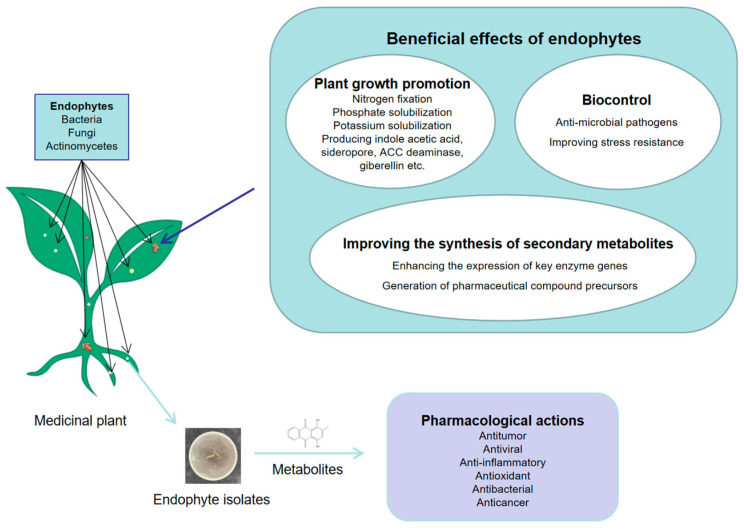
Potential functions of endophytes in medicinal plants.

**Table 1 life-13-01695-t001:** Endophytic bacteria resources isolated from medicinal plants in recent years.

Host Plant	Main Therapeutic Activities of the Host	Major Medicinal Components	Tissues	Endophytic Bacteria	References
*Atractylodes macrocephala* Koidz.	Anti-tumor, anti-inflammatory, anti-aging, anti-oxidant, anti-osteoporotic, neuroprotective, immunomodulatory, and improving gastrointestinal function and gonadal hormone regulation	Atractylenolide I, atractylenolide II, and atractylon	Root, stem, and leaf	*Bacillus* sp., *Rhodococcus* sp. *Mycobacterium* sp., *Pseudomonas* sp., *Mycolicibacterium* sp., *Leucobacter* sp., *Enterobacter* sp., *Rhizobium* sp., *Glutamicibacter* sp., and others, for a total of 58 genera	[15,16]
*Dendrobium officinale*	Anti-oxidant, anti-inflammatory, anti-apoptotic, and anti-cancer	Polysaccharides	Stem	*Bacillus* sp., *Enterobacter* sp., *Klebsiella* sp., *Pantoea* sp., *Pseudomonas* sp., *Curtobacterium* sp., *Burkholderia* sp., *Microbacterium* sp., *Lysinibacillus* sp., and others, for a total of 23 genera	[18,19]
Mulberry	Anti-oxidant, anti-inflammatory, and anti-tumor	Polysaccharides, carotenoids, rutin, resveratrol, and anthocyanins	Stem	*Pantoea* sp., *Bacillus* sp., *Pseudomonas* sp., *Curtobacterium* sp., *Sphingomonas* sp., and others, for a total of 36 genera	[20,21,22,23]
Marigold (*Calendula officinalis* L.)	Anti-inflammatory, anti-cancer, anti-helminthic, anti-diabetes, wound healing, hepatoprotective, and anti-oxidant	Terpenoids, flavonoids, triterpeneol esters, steroids, phenolic compounds, carotenes, triterpenoids, essential oils, quinones, fatty acids, minerals, saponins, carbohydrates, sterols, and tocopherols	Root and shoot	*Pantoea* sp., *Enterobacter* sp., *Pseudomonas* sp., *Achromobacter* sp., *Xanthomonas* sp., *Rathayibacter* sp., *Agrobacterium* sp., *Pseudoxanthomonas* sp., and *Beijerinckia* sp.	[24,33]
*Camellia sinensis*	Anti-obesity, anti-diabetes, and anti-cardiovascular disease	Catechins, theaflavins, tannins, and flavonoids	Leaf and root	*Bacillus* sp., *Acinetobacter* sp., *Stenotrophomonas* sp., *Brevundimonas* sp., *Pseudomonas* sp., *Ochrobactrum* sp., *Alcaligenes* sp., and others, for a total of 16 genera	[25,34]
*Handroanthus impetiginosus*	Anti-inflammatory, anti-cancer, anti-bacterial, and anti-malaria	Quinoid	Leaf	*Bacillus* sp., *Paenibacillus* sp., *Pseudomonas* sp., *Rhizobium* sp., *Rummeliibacillus* sp., and *Methylobacterium* sp.	[26,35]
European plum (*Prunus domestica*)	Anti-oxidant, anti-allergic, and anti-cardiovascular disease	Anthocyanins	Shoot	*Pseudomonas* sp. and *Agrobacterium* sp.	[29,36]
Mint (*Endostemon obtusifolius*)	Anti-inflammatory, anti-bacterial, anti-viral, scolicidal, immunomodulatory, anti-tumor, neuroprotective, anti-fatigue, and anti-oxidant	Menthol, menthone, neomenthol, and iso-menthone	Leaf and root	*Paenibacillus* sp. etc.	[30,37]
*Centella asiatica*	Anti-inflammatory, anti-aging, and anti-oxidant	Triterpenoids, flavonoids, vitamins, tannins, polyphenol, and volatile oils	Leaf	*Pseudomonas* sp., *Novosphingobium* sp., *Chryseobacterium* sp., *Enterobacter* sp., *Agrobacterium* sp., *Pantoea* sp., and *Paraburkholderia* sp.	[31]
*Archidendron pauciflorum*	Antibacterial, antioxidant, antidiabetic, and antihyperlipidemic	Alkaloid, flavonoid, tannin, saponin, glycoside, steroid, and terpenoid	Root, leaf, and stem	*Bacillus* sp. etc.	[32]

**Table 2 life-13-01695-t002:** Endophytic fungal resources isolated from medicinal plants in recent years.

Host Plant	Main Therapeutic Activities of the Host	Major Medicinal Components	Tissues	Endophytic Fungi	References
*Aconitum heterophyllum*	Analgesic, anti-inflammatory, antiarrhythmic, anti-parasitic and anticancer	Tannins, flavonoids, saponins, diterpenes, and alkaloids	Root, stem, and leaf	*Arthrinium* sp., *Chaetomium* sp., *Purpureocillium* sp., *Alternaria* sp., *Penicillium* sp., *Aspergillus* sp., *Cladosporium* sp., *and Bjerkandera* sp.	[38,39]
*Crocus sativus*	Against cardiovascular and Alzheimer disease, anti-oxidant, anti-inflammatory, antitumor, and anti-depressant	Alkaloids, anthocyanins, carotenoids, flavonoid, phenolic, saponins, and terpenoids	Corm, scape, leaf, petal, and stigma	*Penicillium* sp., *Sistotrema* sp., *and Bjerkandera* sp.	[40,41,42]
*Salvia miltiorrhiza*	Anti-inflammatory, antioxidant, anti-thrombotic, and cardio-protective	Salvianolic acid A, salvianolic acid B, lithospermic acid, rosmarinic acid, danshensu, tanshinone I, tanshinone IIA, tanshinone IIB, cryptotanshinone, and dihydrotanshinone I	Root	*Fusarium* sp., *Epicoccum* sp., *Aspergillus* sp., *Arthrinium* sp., *Coprinellus* sp., *Dictyosporium* sp., *Colletotrichum* sp., *Rhizoctonia* sp., *Phomopsis* sp. *and Pithomyces* sp.	[43]
Mint (*Endostemon obtusifolius*)	Anti-inflammatory, anti-bacterial, anti-viral, scolicidal, immunomodulatory, anti-tumor, neuroprotective, anti-fatigue and anti-oxidant	Menthol, menthone, neomenthol and iso-menthone	Root and leaf	*Fusarium* sp. etc.	[30,37]
*Dendrobium officinale*	Anti-oxidant, anti-inflammatory, anti-apoptotic and anti-cancer	Polysaccharides	Leaf	*Colletotrichum* sp., *Fusariumand* sp., and *Trichoderma* sp.	[44]
*Codonopsis pilosula*	Immunomodulatory, antitumor, antioxidant, neuroprotective, antiviral, anti-inflammatory, anti-fatigue, hypoglycemic, anti-hypoxia, renoprotective, gastroprotective, hepatoprotective and prebiotic	Polysaccharides	Root	*Fusarium* sp., *Aspergillus* sp., *Alternaria* sp., *Penicillium* sp., *Plectosphaerella* sp. etc.	[45,46]
*Vernonia anthelmintica*	Anti-vitiligo, anti-diabetic, anti-inflammatory, antipsoriatic, neuroprotective, hepatoprotective, analgesic, antipyretic, anti-oxidant, anti-parasitic, anti-microbial, anti-proliferative and immunomodulatory	Phenolic acids, chalcones, flavonoids, terpenes, fatty acids, steroids, and miscellaneous compounds	Flower	*Ovatospora* sp., *Chaetomium* sp., *Thielavia* sp. *and Aspergillus* sp.	[47,48]

**Table 3 life-13-01695-t003:** Endophytic actinomycete resources isolated from medicinal plants in recent years.

Host Plant	Main Therapeutic Activities of the Host	Major Medicinal Components	Tissues	Endophytic Actinomycetes	References
*Dioscorea opposita*	Phenols, flavonoids, saponins, anthocyanins, carotenoids, allantoins, and polysaccharides	Improving the cardiovascular system, regulating immune function, anti-tumour, anti-bacterial, anti-inflammatory, and anti-diabetic	Root, stem and leaf	*Streptomyces* sp.	[49,50]
*Eucommia ulmoides* Oliv.	Anti-hypertension, anti-diabetes, neuroprotection, anti-cancer, anti-inflammatory, anti-osteoporotic, hepatoprotection and kidney protection	Iridoids, lignans	Root and leaf	*Nocardia* sp.	[51,52,53]
*Thymus roseus*	Anti-inflammatory, anti-bacterial, anti-viral, anti-oxidant, anti-cancer, and anti-thrombus	Terpenes, essential oils	Root, stem and leaf	*Nocardiopsis* sp., *Micrococcus* sp., *Kocuria* sp., and etc.	[54]
*Viola odorata*	Anti-inflammatory, anti-diabetes, anti-cancer, diaphoretic, diuretic, emollient, expectorant, antipyretic and laxative	Saponins, glycoside, mucilage, vitamins, and alkaloids	Root	*Streptomyces* sp.	[55,56]
*Xanthium sibiricum*	Anti-inflammatory, treating asthma, improving immunity	Sesquiterpenoids, lignans, flavonoids, steroids, caffeoylquinic acids and thiazinodiones	Leaf and seed	*Streptomyces* sp.	[57,58]
*Kandelia candel*	Antimicrobial, anti-oxidant	Phenols, flavonoids, anthocyanins and lignins	Root	*Nocardioides* sp.	[60,62,63]
*Mentha haplocalyx*	Anti-microbial, anti-inflammatory, anti-oxidant, anti-tumor, gastrointestinal protective, and hepatoprotective	Polyphenolic acids, flavonoids, monoterpenoids, and glycosides	Bark	*Nakamurella* sp.	[61,64]

## Data Availability

Data are contained within the article.

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
