# Peer review of "Cultivable Endophyte Resources in Medicinal Plants and Effects on Hosts"

_life, 2023, doi:10.3390/life13081695_

Round 1
Reviewer 1 Report
Authors performed a revision about endophytes from medicinal plants. The topic is very interesting andthere is a lot of controversy about the role of endophytes in the biosynthesis of secondary metabolites. I think that the way authors faces this challenge is not good enough. Sections 2.1, 2.2, and 2.3 are just a list of how many endophytes were isolated from a specific medicinal plant resulting in a boring lecture. Authors should add more information in order to be more attractive to the audience. For example, they should mention the main active compounds present in the medicinal plants and how endophytes are involved in their biosynthesis including the mechanisms and the regulation processes. Also, I suggest to add in Tables 1, 2, and 3 the active compounds present in the mentioned plant species. Also, it is important to add a figure where you show or propose the principal ways endophytes can affect the biosynthesis of compounds or the plant development. The above figure can substitute figure 2 which do not contribute to the story. Discussion section is so vague and it must be rewritten considering the most important facts. Also, i consider important to discuss the biotechnological challenges of using endophytes instead plants, because since too many years ago, scientifics had tried to use microorganims to biosynthesize important plant medicinal compounds. Finally, in the attached file, I include some minor comments.

I think the quality of English is enough with some minor points included in the attached file.
Author Response
亲爱的审稿人:
Thank you for your valuable feedback on the revision of this article. We have carefully considered and made significant revisions based on your feedback. The content is as follows:
- We added the methodology of this review at the end of the introduction;
- For the contents and tables in 2.1, 2.2 and 2.3, I have added more introductions about hosts, such as the main efficacy and Active ingredient of medicinal plants, which makes this part more attractive;
- We deleted the original Figure 2 and redrawn a picture about the function of endophytes to more intuitively express the theme of the article;
- We made many modifications to the discussion section. We first want to discuss how to explore more medicinal plant endophytic resources through more advanced technologies and cultivate them. It is not enough to only use high-throughput sequencing to explore endophytes, but more importantly, how to obtain these endophytes, which is very necessary and meaningful. Secondly, taking the production of paclitaxel as an example, we discussed the application, difficulties and solutions of endophytesinstead of medicinal plants to produce bioactivenatural product, hoping to apply this research method to more rare medicinal plants. We made many modifications to the discussion section.
At the same time, we have carefully studied the problems and mistakes you have marked in this article and made corresponding modifications, as shown below:
L27. We changed ‘its plants’ to ‘medicinal plants’.This makes the article more fluent.
L70. We reread the original text and found that its full name is Atractylodes macrocephala Koidz. Thank you for your professionalism and rigor.
L75. On a rereading of the references, the original text describes multiple varieties of the genus Dendrobium that have pharmacological functions, so they use the single word ‘Dendrobium’. We also want to express the same meaning, but we realize that italics should not be used, but only when describing the precise species of ‘Dendrobium officinale’.
L96. We optimized the expression of sentences.
L124. Mistakes in English grammar. We have corrected it.
L152. We have supplemented the full name of IAA with indoleacetic acid.
L178. We have revised the mistake of this sentence.
L183 It was our description that made the mistake. We've changed it to growth promoting ability.
L185. We have corrected the word format
L195. We added a simple and accurate background of salvia miltiorrhiza and tanshinone. This is crucial for subsequent descriptions.
L236.We have corrected the format
L261-264.We summarize the sentences so that they are not wordy.
L284.We have rewritten this part, so this part has been replaced by new content.
All modifications have been marked and placed in the attachment. Thank you again for your patience and dedication.
Best regards!
Xiaoying Cao
29 July 2023

Reviewer 2 Report
This revision aimed to 'provide ideas for improving the quality of medicinal plants, developing more microbial resources, exploring more medicinal natural products, and providing help for developing the research on medicinal plants and endophytes'. In the MS, the authors addressed the diversity of endophytes from medicinal plants, some mechanisms of plant-growth promotion, and their effect on plants, including stimulating the production of secondary metabolites of pharmaceutic importance.
Overall the MS is well organized but needs substantial improvement in the following: 1) Methodology of this review, 2) More context on the examples of plants (botanical families, pharmaceutical activities, crop or wild plants), 3) Mechanisms of action of microbial endophytes and not less critical correct wrong notions of endophytic interactions.
Although the revision is limited to the last five years of research, it covers their objectives sufficiently and allows to detect gaps in the information. Therefore I have suggested that the authors elaborate more on the discussion to increase the impact of this review.
Please, address the following comments and corrections
-In general, the abstract needs polishing
L11-12. Verbose, please rephrase.
L17. It is more appropriate, 'tolerance'
L18. Revise sentence; I guess 'it' refers to 'endophytes'; clarify.
Introduction. Add the Methodology, software, and formal analysis for this review (e.g, number of papers analyzed and the keywords used in the search).
L38. This sentence reads better without 'however,'
L54. Change this expression; it sounds very similar to the title of the cited paper.
L104. Are these dominant taxa known for their therapeutic relevance?
L114. Using uppercase in 'Endophytes', 'Endophytic Fungi', and so on is unnecessary.
L115 and similar. Add the botanical family systematically; note that some families have ancestral applications. Another detail overlooked in the MS is the systematic description of the therapeutic activities of medicinal plants. Both details could be part of tables, thus giving the reader a better association of plants, their microbial associates, and their pharmaceutic properties. In the text, the authors should consistently provide a geographic context of the plants; note that in several cases, plant metabolites are more effectively produced under certain climate conditions and cropped or wild conditions.
The authors may consider adding the following:
Caruso G., Abdelhamid M.T., Kalisz A., Sekara A. Linking endophytic fungi to medicinal plants therapeutic activity. A case study on Asteraceae. Agriculture. 2020;10:286. doi: 10.3390/agriculture10070286
L145. This sentence fits better at the beginning of the previous section about the diversity of endophytes.
L153. Indicate a better indication than 40% culture dilution, or clarify its meaning.
L156. Indole acid?
L159. Correct 'phase solubilization'
L161. What kind of metabolite promotes the growth of Albizia lebbeck?
L162. What does 'many plants' mean? Please clarify.
L63-165. Rephrase for clarity.
L167-169. Miss interpretation. Endophytes establish multiple ecological (biotic) interactions with their hosts; hence they could not be considered environmental factors. Please, modify accordingly.
L173-174. The statement needs revision; the literature has widely addressed this topic. Add at least a few examples of hormones (and or their precursors) with an effect on plant performance.
L176. Heading, 'stress tolerance' is what could be enhanced.
Figure 2 seems out to date. I suggest illustrating new approaches for endophyte culture, e. g. culturomics, as reviewed by Riva et al., 2022 and included in this review.
L210-215. These sentences fit better in the Discussion
L261-264. Repetitive ideas; please summarize.
L283-292.Why are the examples of high-throughput sequencing relevant to the discussion? what is the pharmaceutical importance of those plants?. This review might enhance its impact by discussing extensively the importance of culturomics as an approach to developing more microbial resources, as concluded in L295-299.
Moderate editing of English language is required, see comments and corrections.
Author Response
Dear Reviewer:
Thank you for your valuable feedback on the revision of this article. We have carefully considered and made significant revisions based on your feedback. The content is as follows:
- We added the methodology of this review at the end of the introduction;
- For the contents and tables in 2.1, 2.2 and 2.3, I have added more introductions about hosts, such as the main efficacy and Active ingredient of medicinal plants, which makes this part more attractive;
- We deleted the original Figure 2 and redrawn a picture about the function of endophytes to more intuitively express the theme of the article;
- We made many modifications to the discussion section. We first want to discuss how to explore more medicinal plant endophytic resources through more advanced technologies and cultivate them. It is not enough to only use high-throughput sequencing to explore endophytes, but more importantly, how to obtain these endophytes, which is very necessary and meaningful. Secondly, taking the production of paclitaxel as an example, we discussed the application, difficulties and solutions of endophytesinstead of medicinal plants to produce bioactivenatural product, hoping to apply this research method to more rare medicinal plants. We made many modifications to the discussion section.
We have also studied and revised your opinions one by one, and the specific contents are as follows:
L11-12. Verbose, please rephrase.
Answer:We simplified this part of the abstract.
L17. It is more appropriate, 'tolerance'
Answer:’Tolerance’ is indeed more appropriate.
L18. Revise sentence; I guess 'it' refers to 'endophytes'; clarify.
Answer:The endophyte that ’it’ really refers to, in order to avoid misunderstanding, we changed ‘it’ to ‘endophyte’.
Introduction. Add the Methodology, software, and formal analysis for this review (e.g, number of papers analyzed and the keywords used in the search).
Answer:Methodology is really an important part of the review, and we added it at the end of the introduction.
L38. This sentence reads better without 'however,'
Answer:We deleted 'however’.
L54. Change this expression; it sounds very similar to the title of the cited paper.
Answer:In order to avoid too much duplication with references, we have replaced the ‘treasure house’ with ‘more resources’.
L104. Are these dominant taxa known for their therapeutic relevance?
Answer:Among the isolated strains, the genus Epicoccus sp. SX19 and Colletotrichum gloeosporioides have strong inhibitory effects on five pathogenic bacteria. We have supplemented the content.
L114. Using uppercase in 'Endophytes', 'Endophytic Fungi', and so on is unnecessary.
Answer:We have revised the format of the words. Thank you very much for your rigor.
L115 and similar. Add the botanical family systematically; note that some families have ancestral applications. Another detail overlooked in the MS is the systematic description of the therapeutic activities of medicinal plants. Both details could be part of tables, thus giving the reader a better association of plants, their microbial associates, and their pharmaceutic properties. In the text, the authors should consistently provide a geographic context of the plants; note that in several cases, plant metabolites are more effectively produced under certain climate conditions and cropped or wild conditions.
The authors may consider adding the following:
Caruso G., Abdelhamid M.T., Kalisz A., Sekara A. Linking endophytic fungi to medicinal plants therapeutic activity. A case study on Asteraceae. Agriculture. 2020;10:286. doi: 10.3390/agriculture10070286
Answer:We refer to the literature you provided, add many backgrounds such as pharmacological activities, collection sites and medicinal components of hosts, and supplement the contents of the table, thus increasing the attraction of this part. These factors have also played an important role in the distribution and function of endophytes, which is a suggestion worthy of improvement.
L145. This sentence fits better at the beginning of the previous section about the diversity of endophytes.
Answer:We adjusted the content and position of these words and put the newly changed content at the beginning of the section 2, which is more suitable for the content of the article.
L153. Indicate a better indication than 40% culture dilution, or clarify its meaning.
Answer:We apologize for not clarifying the content of the original author's experiment. The concentration of 40% culture solution is the optimum concentration after the bacteriostatic test with three concentration gradients established by the original author. We have presented the content of the experiment in a concise way.
L156. Indole acid?
Answer:We have supplemented the full name of IAA with indoleacetic acid.
L159. Correct 'phase solubilization'
Answer:Very serious single error, we have changed it to ‘phosphate solubilization’.
L161. What kind of metabolite promotes the growth of Albizia lebbeck?
Answer:Once again, we apologize for not explicitly proposing specific compounds, because ‘Gibberellin’ can promote growth.
L162. What does 'many plants' mean? Please clarify.
Answer:’Many plant’ refers to ‘wheat, barley and millet’. At first, we thought that these crops were not medicinal plants, but we really couldn't ignore the work of researchers, so we made the experiment more clear.
L63-165. Rephrase for clarity.
Answer:We reorganized the language to make the description clearer and more concise.
L167-169. Miss interpretation. Endophytes establish multiple ecological (biotic) interactions with their hosts; hence they could not be considered environmental factors. Please, modify accordingly.
Answer:We have modified the content, and endophyte is an important ‘factor’ affecting the growth of medicinal plants, not ‘environmental factors’.
L173-174. The statement needs revision; the literature has widely addressed this topic. Add at least a few examples of hormones (and or their precursors) with an effect on plant performance.
Answer:It is very important to add some examples. We added some plant growth hormones secreted by endophytes, thus making the article more credible.
L176. Heading, 'stress tolerance' is what could be enhanced.
Answer:We have changed it to 'stress tolerance' .
Figure 2 seems out to date. I suggest illustrating new approaches for endophyte culture, e. g. culturomics, as reviewed by Riva et al., 2022 and included in this review.
Answer:We think that the cultivation method of endophytes is extremely important to obtain more microbial resources, but we think that figure2 is not helpful and outdated, so we decided to delete the original picture and redraw a picture about the function of endophytes in medicinal plants to show the content of the article more clearly.
L210-215. These sentences fit better in the Discussion
Answer:We supplemented the content of using endophytes instead of rare medicinal plants to produce active natural products, and discussed the production of endophytic fungi and paclitaxel in ‘Taxus chinensis’ as examples. This is a very meaningful research to be used in drug development and to protect and cherish medicinal plant resources.We studied the literature you shared, and also found some literature about endophyte culture, and discussed with high-throughput sequencing whether we can combine microbial culture technology and high-throughput sequencing technology to obtain more microbial resources.
L261-264. Repetitive ideas; please summarize.
Answer:We summarize the sentences so that they are not wordy.
L283-292.Why are the examples of high-throughput sequencing relevant to the discussion? what is the pharmaceutical importance of those plants?. This review might enhance its impact by discussing extensively the importance of culturomics as an approach to developing more microbial resources, as concluded in L295-299.
Answer:What we want to discuss about high-throughput sequencing is to prove that there are a lot of microbial resources in medicinal plants, but we can only prove their existence, but we can't separate them. The function and growth environment of medicinal plants also affect the function and richness of endophytes, so we discuss the new culture and separation methods of endophytes mentioned before, in order to obtain more bioactive microbial resources. We think this is a far-reaching work, which is worthy of further research by researchers.
All modifications have been marked and placed in the attachment. Thank you again for your patience and dedication.
Best regards!
Xiaoying Cao
29 July 2023

Round 2
Reviewer 2 Report
Dear Editor,
The corrections and modifications by the authors have improved the MS. Now, it has more direction, highlighting at the same time the therapeutic effects of medicinal plants and the role of their endophytic associates. As I requested, methods are now included, tables and figures have more context, and the discussion is more focused on current gaps in knowledge on endophytes, thus increasing the relevance of this paper. Therefore I suggest the acceptance of the MS.